# Low Adenovirus Vaccine Doses Administered to Skin Using Microneedle Patches Induce Better Functional Antibody Immunogenicity as Compared to Systemic Injection

**DOI:** 10.3390/vaccines9030299

**Published:** 2021-03-22

**Authors:** Olivia Flynn, Kate Dillane, Juliane Sousa Lanza, Jennifer M. Marshall, Jing Jin, Sarah E. Silk, Simon J. Draper, Anne C. Moore

**Affiliations:** 1School of Pharmacy, University College Cork, T12 XF62 Cork, Ireland; olivia.flynn@cit.ie (O.F.); kdillane@ucc.ie (K.D.); julianelanza@gmail.com (J.S.L.); 2The Jenner Institute, University of Oxford, Oxford OX3 7DQ, UK; jennifer.marshall@ndm.ox.ac.uk (J.M.M.); jing.jin@ndm.ox.ac.uk (J.J.); sarah.silk@ndm.ox.ac.uk (S.E.S.); simon.draper@ndm.ox.ac.uk (S.J.D.); 3School of Biochemistry and Cell Biology, University College Cork, T12 XF62 Cork, Ireland

**Keywords:** adenovirus, virus vector vaccine, microneedle, skin, dose, stability, antibody, anti-vector immunity

## Abstract

Adenovirus-based vaccines are demonstrating promising clinical potential for multiple infectious diseases, including COVID-19. However, the immunogenicity of the vector itself decreases its effectiveness as a boosting vaccine due to the induction of strong anti-vector neutralizing immunity. Here we determined how dissolvable microneedle patches (DMN) for skin immunization can overcome this issue, using a clinically-relevant adenovirus-based *Plasmodium falciparum* malaria vaccine, AdHu5–PfRH5, in mice. Incorporation of vaccine into patches significantly enhanced its thermostability compared to the liquid form. Conventional high dose repeated immunization by the intramuscular (IM) route induced low antigen-specific IgG titres and high anti-vector immunity. A low priming dose of vaccine, by the IM route, but more so using DMN patches, induced the most efficacious immune responses, assessed by parasite growth inhibitory activity (GIA) assays. Administration of low dose AdHu5–PfRH5 using patches to the skin, boosted by high dose IM, induced the highest antigen-specific serum IgG response after boosting, the greatest skewing of the antibody response towards the antigen and away from the vector, and the highest efficacy. This study therefore demonstrates that repeated use of the same adenovirus vaccine can be highly immunogenic towards the transgene if a low dose is used to prime the response. It also provides a method of stabilizing adenovirus vaccine, in easy-to-administer dissolvable microneedle patches, permitting storage and distribution out of cold chain.

## 1. Introduction

The high cost of maintaining vaccine cold chains has been identified as a key challenge that must be overcome to achieve global immunization goals. Live and whole inactivated viruses remain a sizeable proportion of clinically used vaccines. Recombinant virus vectors, and in particular, replication incompetent, recombinant adenoviruses are leading vaccine candidates for old, emerging, and pandemic diseases, including several candidates to protect against SARS-CoV-2-induced COVID-19 [1,2,3]. These vaccines are heat sensitive and must be stored and distributed at low temperatures. Stabilization of live virus vaccines is challenging, but overcoming this issue is a critical factor for the successful and equitable deployment of vaccines in the field [4].

Induction of immunity to the adenovirus vector severely limits its repeated use due to immune-mediated neutralization of the re-administered virus. Clinically, this is being addressed through the use of heterologous prime-boost immunization, where a different vaccine platform, such as the poxvirus vector MVA (Modified Vaccinia Virus Ankara) [5,6,7,8] or subunit, adjuvanted protein [9] is used in the second and subsequent immunizations, or through the use of serologically different adenoviruses [10,11]. Although strong antigen-specific immunity can be induced in clinical trials, the costs and logistics of manufacturing and distributing two or more vaccines and administering them in the correct sequence in every individual, would need to be economically justified before widespread implementation, or alternatively only utilized in epidemic outbreaks.

We previously demonstrated that adenovirus vaccine administration using silicon microneedles permitted repeated use of the same adenovirus vaccine due to the induction of significantly lower anti-vector antibody responses compared to injection with a needle and syringe in mice [12]. However, although silicon microneedles overcome some logistic issues, they do not address vaccine stability. Dissolvable microneedles are micron-scale protrusions that are sharp enough to insert into the skin. They are composed of suitable excipients that incorporate, and should stabilize, vaccines in a dry, solid state. On skin insertion the microneedle material dissolves and the vaccine is delivered to the body. Dissolvable microneedle (DMN) patches for skin-based immunization offer several logistic and economic benefits to healthcare systems and individual end-users, including: ease of use; lack of requirement for vaccine reconstitution; more efficient and cost-effective logistics, by removal of cold chain; the elimination of needles, syringes, and hazardous waste; and small package size. We previously developed a novel, drop-dispensing based DMN patch production method that overcomes the waste associated with conventional methods of dissolvable microneedle production [13,14,15]. We also developed suitable formulations for stabilizing vaccines in DMN patches to preserve vaccine integrity during microneedle fabrication, upon drying and during subsequent storage [16].

Dose sparing has been suggested as a key benefit of microneedle-mediated vaccine administration, however whether this translates to humans or larger animals has yet to be determined. A small number of published clinical studies have assessed the impact of adenovirus-based vaccine dose on T cell responses [17,18] and antibody titers [19,20,21] using conventional intramuscular delivery. We therefore wanted to determine how adenovirus vaccine dose impacts on the induction of antibody responses when the vaccine is administered by the intramuscular (IM) or patch-mediated skin route. Here, we chose human adenovirus serotype 5 (AdHu5) as a model adenovirus without intellectual property restrictions, and which is also suitable for thermostability and administration route studies. The immunogenicity and efficacy of AdHu5 vaccines can be readily matched by clinically-applicable simian adenoviruses, as previously demonstrated [22,23].

To date, the administration of adenovirus vaccines by microneedle patches in mice has focused on the induction of T cell responses to HIV antigens [24,25], and there has been no understanding of the effect of anti-vector immunity, or the induction of functional/protective humoral immunity using DMN patches in any model. In this study, we address this problem and examine the induction of vector-specific and functional antigen-specific antibody responses using DMN patches fabricated using our simple process.

The development of highly effective and durable vaccines against the human malaria parasites *Plasmodium falciparum* and *P. vivax* remains challenging [5,26]. One vaccine strategy is targeted at the blood-stage of the disease, by blocking merozoite invasion of red blood cells. This approach will likely require the maintenance of high titres of neutralizing antibodies to merozoite proteins. *Plasmodium falciparum* reticulocyte-binding protein homolog 5 (RH5) has emerged as a strong blood-stage vaccine candidate, and is being clinically tested. In parallel with this development, non-human primate studies have demonstrated that the in vitro assay of antibody-mediated growth inhibition activity (GIA) against *P. falciparum* serves as an in vitro correlate of vaccine-induced in vivo protection when using vaccines targeting PfRH5 and other blood-stage antigens [27,28,29].

The objective of this study was therefore to determine how the dose and method of delivery, either IM injection or skin administration using a DMN patch, of an adenovirus vectored blood-stage malaria vaccine impacts on the induction of neutralizing anti-parasitic antibodies and anti-vector antibodies in mice.

## 2. Methods

### 2.1. Vaccine

Replication-deficient adenovirus human serotype 5 (AdHu5) expressing full-length *Plasmodium falciparum* RH5 (AdHu5–PfRH5) was previously described [27,30]. Antigens were cloned into a replication-deficient AdHu5 genome downstream of a mammalian secretory signal (from human tissue plasminogen activator), and viruses were prepared using previously described techniques for AdHu5. All doses of vaccine are expressed as live, infectious units (ifu) but where relevant in the text, the dose measured in viral particles (vp) is also reported.

### 2.2. Dissolvable Microneedle Fabrication

Pyramidal-shaped silicon microneedles consisting of 5 × 5 array (length: 500 μm, base diameter: 333 μm, Tyndall National Institute) were used as a molding template for fabrication of microneedle cavities in polydimethylsiloxane (PDMS) molds, as previously described [15,31,32]. The vaccine was formulated with a 15% (*v*/*v*) trehalose-based solution for the production of dissolvable microneedle patches. Approximately 7 μL of formulation was delivered directly into 25 microneedle pores of a 1 cm [2] mold. Following delivery of the formulation onto molds, microneedles were dried for 18 h at room temperature (approximately 21 °C) in the presence of desiccant. After transfer onto medical grade adhesive tape (1525 L Poly Med tape, 3 M) arrays were kept in a sealed container in a dry ambient environment in the presence of desiccant until use.

### 2.3. Stability Study

DMN patches containing AdHu5–PfRH5 with a dose of 1 × 10^7^ ifu per 1 cm^2^ patch were sealed into nitrogen-purged individual glass vials with desiccant. Packaged and sealed vaccine-loaded DMN were stored in a stability chamber at 25 °C and 60% relative humidity (RH), or 40 °C and 75% RH, for the indicated time, according to ICH Q1A guidelines [33] (*n* = 6 per group per timepoint, except at 6 months when *n* = 4 per group). Samples were dissolved in 32 μL complete RPMI media, and viable AdHu5–PfRH5 was quantified using a hexon immunoassay (QuickTiter Immunoassay, Cambridge Bioscience, Cambridge, UK).

### 2.4. Immunization

Female Balb/c mice, 4–6 weeks old (Harlan, Hillcrest, UK), were used in all experiments which were conducted in accordance with the terms of licenses from the Irish Department of Health and Children under the Cruelty to Animals Act (license number B100/4034) and according to the approval of the UCC AECC Committee. Mice were immunized with the dose of vaccine indicated in the text and figure legends. For each patch administration, mice were pre-anesthetized with ketamine hydrochloride and medetomidine hydrochloride, and the patches were applied to the dorsal surface of each ear, pressing the fingers against the animals’ skin for 10 s. The ear was then loosely wrapped with Elastoplast to permit the maintenance of the patches on the skin. An anesthetic reversal agent, atipamezole (anti-sedan), was then administered to mice. Patches were in place overnight as previously outlined [34]. Alternatively, vaccine was administered with a conventional 28 G needle and syringe intramuscularly (IM) into the anterior thigh of the hind leg (50 μL per mouse divided across both legs). Mice were primed on day 0 and boosted by the IM route or using a microneedle patch at day 54, as outlined in the figures and text.

### 2.5. Immunogenicity Analysis

Antibody responses to the vector and to the antigen were determined by ELISA as previously described [12,35,36], using 100 ng/well of purified recombinant PfRH5 antigen (called “RH5.1”) produced from a *Drosophila melanogaster* Schneider-2 (S2) stable cell line [37] for antigen-specific responses and AdHu5-mCherry (3 × 10^8^ vp/well in PBS) for vector-specific responses [12]. Endpoint titres were taken as the *x*-axis intercept of the dilution curve at an absorbance value 3× standard deviations greater than the OD_595_ for naïve mouse serum (typical cut-off OD_595_ for positive sera = 0.1). A standard serum sample was included in all assays as a reference control.

### 2.6. Functional Antibody Efficacy Analysis: Growth Inhibitory Activity Assay

The ability of antibodies to inhibit growth of *P. falciparum* 3D7 clone parasites in vitro was assessed by a standardized GIA assay using purified IgG at 1 mg/mL, as previously described in detail [38,39]. Briefly, purified IgGs from serum samples were pre-adsorbed with uninfected human O+ erythrocytes to remove any anti-human erythrocyte immunoglobulins, sterile filtered, and heat inactivated. The GIA assay was performed on these samples using human erythrocytes parasitized with late trophozoite and schizont stages of *P. falciparum*. Parasite growth after 40 h of culture was determined by a biochemical assay specific for parasite lactate dehydrogenase, and the results were determined by the OD650. Values obtained with test IgGs were compared with those obtained with parasites incubated with a pool of malaria-naïve human control serum and with uninfected red cells, and are reported as % GIA in the test sample versus control.

## 3. Results

### 3.1. Long-Term Vaccine Stabilisation Outside Cold Chain Conditions

We previously demonstrated the short-term stability of an AdHu5 vector embedded in our DMN patches and kept at ambient temperature over 14 days [15]. Here, we stressed the vaccine at ambient and high temperature conditions for 6 months to better determine the vaccine’s stability in dissolvable microneedle patches. AdHu5–PfRH5 vaccine-containing DMN patches were stored at 25 °C and 60% relative humidity (RH), or 40 °C and 75% RH, for up to 6 months, according to ICH Q1A guidelines [33]. Adenovirus infectivity was assessed in patches or in liquid at 1 week or 1, 2, 3, and 6 months. Adenovirus stored in the liquid form at the higher temperature, quickly degraded within a week, and viable virus was undetectable at one month (Figure 1A). In contrast, no degradation of the vaccine in patches was observed at 40 °C for up to 2 months, however vaccine degradation was observed when patches were stored at 40 °C for 3 months. No visible changes to the microneedles were observed in these patches stored at 40 °C with up to 6 months storage (Figure 1C). At the accelerated temperature of 25 °C, adenovirus vaccine incorporated into patches was fully stabilized; no loss in viable vaccine (Figure 1A) or visible changes (Figure 1B) could be detected at 6 months. These results demonstrate the significant improvement to the stability of this adenovirus vaccine by incorporation into patches.

### 3.2. Low Dose Skin Administration Significantly Enhances Antigen-Specific Antibody Responses Induced by Repeated Adenovirus Vaccination

We previously demonstrated influenza vaccine dose sparing using dissolvable microneedles in mice [15]. As adenovirus is a live vaccine where one virus unit can produce an unknown antigen dose in situ, we aimed to determine if dose sparing would be relevant to adenovirus vaccines. To address this question, a range of doses of adenovirus expressing *P. falciparum* RH5 antigen, AdHu5–PfRH5 [30], was administered by patches or in liquid format by the IM route to Balb/c mice (Appendix A). A 1 × 10^8^ infectious units, ifu (1.6 × 10^9^ virus particles, vp) vaccine dose by the IM route represented the dose and route of immunization conventionally administered to mice [40]. Only the lowest doses of vaccine by the IM or patch route induced a significantly lower IgG response to the PfRH5 antigen, compared to the conventional vaccine dose (1 × 10^8^ ifu), 8 weeks after the primary immunization (Figure 2A; light grey bars). With respect to the induction of anti-vector immunity, lower vaccine doses by either IM (≤1 × 10^5^ ifu) or skin (≤5 × 10^5^ ifu) route, also induced significantly lower anti-adenovirus IgG responses after the first immunization, compared to the standard 1 × 10^8^ ifu IM immunization (Figure 2B).

All groups were boosted by the same IM route and dose, so that post-boost immunity only reflected differences induced in the primary immunization. All groups had a significantly higher IgG to PfRH5 post-boost, compared to the primary immunization (*p* < 0.05, paired *t*-test) (Figure 2A). Surprisingly, mice primed with lower doses of vaccine (5 × 10^5^ ifu and 5 × 10^4^ ifu) using DMN patches, but not by the IM route, induced consistent and significantly higher antigen-specific IgG responses compared to the conventional IM immunization (Figure 2A). In contrast, a high level of variability was observed in animals primed with low doses of vaccine by the IM route. The IM high dose boost caused a significant increase in anti-vector IgG in all groups compared to the post-prime anti-vector response (Figure 2B). A dose response effect was observed with respect to anti-vector responses post-boost (Figure 2B). Priming doses less than 1 × 10^7^ ifu by either route resulted in significantly lower post-boost anti-vector responses compared to the 1 × 10^8^ ifu IM group.

To determine how anti-AdHu5 IgG antibodies reflected functional, adenovirus neutralizing capacity, titres of neutralizing antibodies (NAb) were determined by hexon infectivity assay for high and low vaccine doses by the IM and skin routes after boosting (Figure 2D). A dose dependence was observed, irrespective of route, whereby low vaccine doses (1–5 × 10^4^ ifu) induced significantly weaker anti-adenovirus NAb titres as compared to priming with 1–5 × 10^7^ ifu or with 1 × 10^8^ ifu IM.

Using the antibody titres to the antigen (Figure 2A) and to the vector (Figure 2B), we analyzed the skewing of the antibody response to the encoded antigen, and away from the virus vector. Conventional immunization, using 1 × 10^8^ ifu AdHu5–PfRH5 by the IM route primed a humoral response, whereby the anti-transgene response was 3.5 times higher than the anti-vector antibody response, as evidenced by an equivalent or higher relative level of vector-specific antibodies compared to antigen-specific antibodies (geometric mean; 3.5, 95% CI; 1.5 to 9.2) (Figure 2C). Lowering the priming vaccine dose resulted in increased ratios of anti-PfRH5 to anti-vector responses, irrespective of route, compared to a single, conventional IM immunization. Boosting mice that were primed with 1 × 10^8^ ifu or 1 × 10^7^ ifu of vaccine by the IM route pushed the ratio of antibodies significantly further towards an anti-vector response, as evidenced by significant decreases in the ratio of anti-PfRH5 antibodies to anti-vector post-boost compared to post-prime. For example, in the IM 1 × 10^7^ ifu group, the post-prime geometric mean ratio was 154 (95% CI: 49 to 488) and post-boost this dropped to a geometric mean ratio of 6 (95% CI: 3 to 11). There was no significant change in the ratio of antigen- to vector-specific antibodies from prime to boost in the lower doses (1 × 10^5^ or 1 × 10^4^ ifu) IM groups. Therefore at lower doses given IM, the ratio of antigen to vector specific antibodies could not be modulated by the boost. In contrast, at the lowest vaccine dose administered by DMN patches, the ratio of antigen to vector specific responses was significantly altered, and increased, from prime to boost; the geometric mean ratio post-prime was 198 (95% CI: 131 to 299) and post-boost this significantly increased to a geometric mean ratio of 1232 (95% CI: 266 to 5713). This regimen of the lowest dose administered with a patch (DMN 5 × 10^4^ ifu) and boosted with a vaccine dose of 1 × 10^8^ ifu by the IM route induced the highest ratio of antigen-specific to vector-specific antibodies; these were significantly higher than the IM 1 × 10^8^ ifu approach, and this was the only prime-boost approach that led to a significant increase in this ratio post-boost. Overall, these results demonstrate that repeated IM immunization favors the expansion of anti-vector immune responses, whereas priming with adenovirus vaccine in DMN patches favors the enrichment of antigen-specific IgG. Priming with low vaccine doses, particularly with a DMN patch, thus favors a further skew towards antigen-specific immunity.

### 3.3. Homologous Dose and Route in the Boosting Immunization Can Enhance Antigen-Specific Antibody Responses

The simplest vaccination regime would be to re-administer the same vaccine dose using the same route in second and subsequent immunizations. However this must be balanced by the induction of strong immunity that provides protection against pathogen challenge. Mice were thus immunized twice with a low (5 × 10^4^ ifu) or high dose (5 × 10^7^ ifu) of AdHu5–PfRH5 by the IM route or with a patch (Figure 3). After the primary immunization, the antibody responses to PfRH5 and to the adenovirus vector demonstrated the same responses as described above. After boosting, mice primed with low dose patches, and boosted with high or low doses patches (DMNlo/DMNlo and DMNlo/DMNhi) had equivalent post-boost anti-PfRH5 antibody responses to repeated high dose IM immunization (Figure 3A). Only high doses of vaccine, by either route, had anti-vector antibodies (Figure 3B). Both of the low dose DMN groups exhibited significantly higher ratios of anti-PfRH5 to anti-vector antibodies after the first immunization, compared to high dose IM immunization (*p* < 0.05) (Figure 3C). After the boost, all groups, including the group receiving two high doses in a patch, exhibited significantly higher ratios of anti-PfRH5 to anti-vector antibodies compared to repeated high dose IM immunization. This highlights that mice primed with low vaccine doses had post-boost responses dominated by anti-PfRH5 antibodies, as relative to anti-vector antibodies.

These results demonstrate that priming and boosting with a low dose (5 × 10^4^ ifu) of vaccine, using patches or by the IM route, significantly skews the antibody response towards the antigen and away from the vector, compared to high dose administration.

### 3.4. Low Dose Priming Induces Higher Levels of Functional Antibodies

We tested the ability of IgG purified from serum to neutralize *P. falciparum* parasites, clone 3D7, in a GIA assay [38,39]. In the first study, serum from mice boosted with high dose vaccine by the IM route (1 × 10^8^ ifu; “IM 10^8”), were evaluated (Figure 4A). Repeated high dose IM immunization induced an antibody response that contained low levels of growth inhibitory antibodies (median 45%, 95% CI; 37% to 56%). A high dose of vaccine administered by patch in the prime and boosted with high dose IM (DMN hi/IM 10^8^) did not significantly increase this response. In contrast, priming with a low dose of vaccine, by the IM route or DMN patches and boosting with 1 × 10^8^ ifu IM induced high levels of antibodies in serum that inhibited parasite growth. GIA levels in the patch primed group (median, 83%, 95% CI: 80–85%) were significantly higher compared to the control, IM, immunized group.

In a second study, we examined the functional efficacy induced by repeated low dose immunization and repeated use of DMN patches (Figure 4B). This study confirmed that administration of a low dose of vaccine by the IM or the patch route primed for a high level of growth inhibitory, neutralizing antibodies. Boosting with a low or high vaccine dose with a patch resulted in GIA levels that were equivalent to repeated low vaccine doses by the IM route. In contrast, repeated administration of a high dose using a patch in both immunizations (DMN hi/DMN hi) induced lower levels of GIA.

Overall, these studies demonstrated that repeated administration of a low vaccine dose by either IM or patch route induced substantially higher GIA compared to the conventional repeated high dose IM regime (IM, 1 × 10^8^ ifu). Low dose priming with a patch and heterologous boosting, using a high dose by the IM route (Figure 4A), induced the most robust and highest level of parasite growth inhibitory antibodies. This DMN lo/IM 10^8 regimen also induced the highest ratio of antigen-specific to vector-specific antibodies (Figure 2C). Therefore, there is a balance between inducing the highest antigen-specific immunity (and functional antibodies) and using the simplest, homologous immunization regime.

### 3.5. Modulation of IgG Subtypes by Altering the Dose and Route of AdHu5–PfRH5

We examined the antigen-specific IgG1 and IgG2a titres to determine if there were significant differences in the ratio of these subtypes due to the vaccine dose and route of immunization. Only groups that had high levels of GIA were examined and compared to the conventional, high dose repeated IM immunization regimen. It was observed that IMlo/IM 10^8 and DMNlo/IM 10^8 significantly increased IgG1 titres compared to repeated high dose IM (Figure 5A). A repeated low dose of vaccine administered with patches (DMNlo/DMNlo) induced significantly reduced antigen-specific IgG2a compared to repeated high dose vaccine by the IM route. Using a patch in the prime skewed the response away from IgG2a; the ratio of IgG1:IgG2a was approximately 100-fold higher when vaccine was administered using a patch in the prime and boost (Figure 5B).

### 3.6. Kinetics of the Antibody Response to the Antigen and to the Vector Induced by the Different Routes and Doses

Finally, we assessed the kinetics of the IgG response over time after a single immunization with a high (1 × 10^8^ ifu) or low (5 × 10^4^ ifu) dose of vaccine, administered by IM injection or DMN patch (Figure 6). The antibody responses to PfRH5 remained relatively stable in most groups over time; the exceptions being a significant increase in the response in the DMN-lo group at day 56, and a significant decrease in titres in the IM-hi group at day 119 (Figure 6A). Significant increases in anti-AdHu5 IgG were observed in both high dose groups at day 56 compared to earlier time points; the magnitude of this antibody response continued to significantly increase to day 119. Anti-AdHu5 IgG responses could be detected in 1 or 2 mice per group in the low dose groups at day 119. No differences were observed in the kinetics of the antibody responses, to PfRH5 or to AdHu5, when a different route was used to administer a high or low dose of vaccine.

## 4. Discussion

Adenovirus vectored vaccines have become a leading platform technology that is demonstrating potential across multiple infectious disease indications. For COVID-19, clinical efficacy has been demonstrated [41], and a number of adenovirus-based COVID-19 vaccines are now being globally deployed. The ability to repeatedly administer the same adenovirus serotype to significantly increase the anti-transgene immunogenicity would greatly improve immunization programs that utilize this technology. However, the issue of anti-vector immunity [1,42] decreases the potency of adenovirus-based vaccines upon repeated systemic use, and especially within a short timeframe. Several strategies have been developed to prevent or overcome this issue, including the use of a heterologous boost, with for example, an MVA-based vector or a protein subunit boost [5,6,7,8,9,19,43,44]. Our study provides a complementary approach of using skin-based delivery with dissolvable microneedle patches to significantly skew the humoral response towards the antigen and away from the vector. We determined that administering lower doses of vaccine skewed the antibody profile towards a higher proportion of antigen-specific, compared to anti-vector, antibodies in mice. Furthermore, stabilizing and administering low doses of vaccine in dissolvable microneedle patches induced the most consistent and favorable skew in the magnitude of antigen-specific IgG, compared to IM injection of liquid vaccine. Finally, immunization with a low vaccine dose with skin patches and boosting with a high dose by the IM route consistently induced the highest level of functional GIA antibodies against the malaria parasite. In contrast, priming with high adenovirus vaccine doses, by either route, induced poor GIA responses.

Dissolvable microneedle technologies are increasingly demonstrating potential as successful vaccine stabilization and delivery systems [45], and have been tested in phase I clinical trials [46,47,48,49]. We previously outlined a novel process for producing vaccine-based dissolvable microneedle patches, which does not result in any waste of precious formulated vaccine [15]. We also demonstrated the short-term stability of an adenovirus virus vector in DMN patches [15]. Here, we extended this finding to demonstrate the significant stability advantages of incorporating adenovirus vectored vaccine in these patches. Previous attempts to stabilize adenovirus vectors in a solid format have involved drying it onto filter membranes [50] or lyophilization [51]. However, decreases in vaccine viability and potency were observed when adenovirus was dried onto filter paper after 4 weeks at 37 °C and 45 °C, and secondly, vaccine on these filters or lyophilized still requires reconstitution and the presence of needles and syringes in order to be administered. In contrast, adenovirus vaccine in our DMN patches retained full potency at 40 °C for somewhere between 8 to 12 weeks, and remained fully viable in this easy-to-administer, needle-free delivery system, at 25 °C for at least 6 months, when this study completed. A thin film format of adenovirus has recently been described that demonstrated a similar stability to what we observed in DMN patches, however the immunogenicity of an adenovirus vaccine in this film is unknown [52]. This format has similar attributes to our patches with respect to ease of administration. To our knowledge, our study is the first demonstration that a live vaccine can be stabilized to this extent in a microneedle patch format.

We previously examined the induction of immunity by adenovirus-based vaccines delivered with a silicon microneedle technology [15,35,53,54]. Other studies have focused on the induction of T cell responses using long (800 μm) dissolvable microneedles fabricated from carboxymethylcellulose (CMC) [24], and only after a single immunization. In a small number of macaques, two adenovirus-based HIV vaccines were coated onto dissolving poly(lactic-co-glycolic acid) (PLGA) microneedles and administered twice, 12 weeks apart. T cell responses were equivalent in microneedle (MN)-treated and IM injected animals [25]. Humoral responses induced after the primary immunization increased after the second MN immunization, however it is unclear how strong this boosting effect was in comparison to IM delivery. Therefore it is possible to boost weak antigen-specific serum antibody titres in macaques by repeated Ad-DMN immunization, however the comparative magnitude of this effect with injected vaccine is unknown [25]. Our present study suggests that repeated low doses, or a low/high prime/boost, repeated adenovirus regime, could be a more suitable regimen to further increase the magnitude of these responses. We previously demonstrated that these sugar-based microneedles fully dissolve in <10 min when administered to porcine skin [15], and therefore deliver vaccine to the body in a short timeframe, similar to the time period for IM delivery. We observed a similar kinetic profile of the antibody response after a single immunization using an IM injection or a DMN patch (Figure 6). Therefore, although the magnitude of the antibody responses were dependent on the dose and delivery system, the kinetic profile of the serum antibody responses were independent of the vaccine format and route.

Clinical trials have assessed the induction of immune responses by repeated adenovirus vector based vaccine administration. For example, repeated IM use of an adenovirus type 35 by the IM route [18] demonstrated that repeated high dose immunization induced higher CD8+ T cells to the encoded antigens, however the phenotype of the boosted CD8+ T cell response was less polyfunctional compared to a single immunization. This suggests that repeated higher doses may not be as effective at inducing protective CD8+ T cells. In malaria vaccine trials, a second immunization with two AdHu5-based vaccines, each at 1 × 10^10^ virus particles, administered concurrently, did not boost antibody responses [55]. In a phase I clinical trial of heterologous prime-boost regime for an Ebola virus vaccine, administering the lowest dose of a chimpanzee adenovirus (ChAd3, 1 × 10^10^ vp) primed for a substantially stronger IgG and neutralizing antibody response subsequent to an MVA boost, compared to priming with 2.5 × 10^10^ vp or 5 × 10^10^ vp of ChAd3, although the low number of volunteers in each group prevents statistical analysis [6]. Repeated use of an adenovirus-based COVID-19 vaccine, namely ChAdOx1-nCoV-19 [56], or malaria vaccine, namely ChAd3-METRAP [57], did not boot T cell responses in either study, but was demonstrated to be capable of increasing antibody titre and modulating antibody effector function to the spike antigen of SARS-CoV2 [56], but not to the malaria antigen [58]; however robust statistical analyses was not feasible in the former study. It is well recognized that T cell receptor (TCR) stimulation strength affects the phenotype of T cell responses; including the generation and function of follicular helper T cells, which are critical to modulating the induction of B cell responses [57]. High antigen dose stimulates T cells of both high and low avidity, thereby permitting expansion of T cells with lower functional quality [57]. Furthermore, increasing antigen dose results in higher expression on T cells of inhibitory receptors, such as PD-1 and CTLA-4 [59]. Here, we add further to these data by demonstrating that a low, compared to high, adenovirus vaccine dose resulted in a significant skew in antibody responses towards the encoded transgene and away from the vector. Decreasing the dose of total adenovirus vector proteins at The effect of the TCR stimulation strength on T cell responses is likely one component of why lower doses of vaccine were more immunogenic in our study, and potentially leads to better priming in the clinicthe time of administration also likely reduces the opportunity for B and T cells to be exposed to vector antigens, while transgene antigen production is maintained above a critical threshold. Finally, decreasing the dose likely also reduced the induction of innate immune response genes, which have been shown to inversely correlate with antigen expression from adenovirus vectored vaccines [60].

## 5. Conclusions

Our findings demonstrate that priming with a low dose of adenovirus vaccine using a skin patch and performing a high dose IM boost with the same vaccine induces the highest level of functional, parasite growth inhibitory antibodies and low anti-vector responses. Fast, cost-effective deployment of vaccines is a required attribute in a vaccine-based response to disease outbreaks. Distribution in regular, room temperature drug distribution networks would address the current cold chain constraints in vaccine deployment. We have developed a stabilized vaccine platform that addresses this issue. Secondly, our skin patch system also addresses the constraints surrounding training vaccinators to prepare and inject vaccines, and safely dispose of biohazardous waste. In addition to these solutions to vaccine access issues, we demonstrate here that this patch system is also capable of significantly enhancing the induction of protective immune responses compared to conventional injection-based immunization. We conclude that further translation of this technology is warranted based on these findings and on the clinical need to address these access and efficacy issues.

## Figures and Tables

**Figure 1 vaccines-09-00299-f001:**
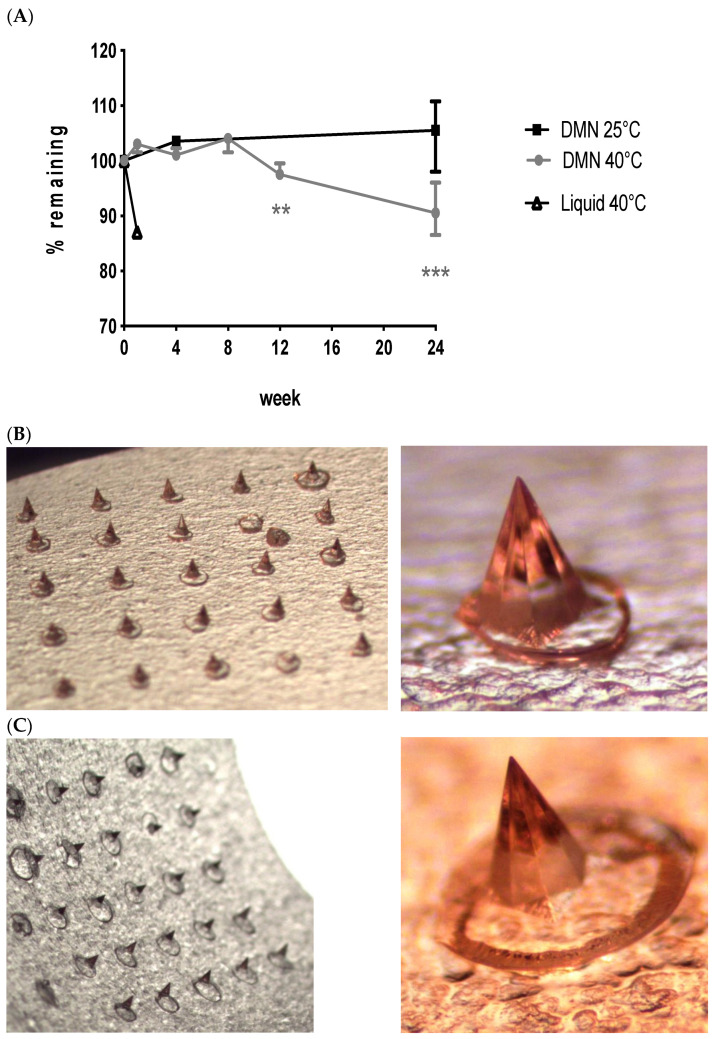
Incorporation of adenovirus vaccine into dissolvable microneedle (DMN) patches significantly stabilized the vaccine for up to 6 months. AdHu5–PfRH5 containing patches were stored in a stability chamber at 25 °C or 40 °C, with the surrounding chamber maintained at 60% or 75% relative humidity (RH), for the indicated time. (**A**) The initial vaccine dose was 1 × 10^7^ ifu per 1 cm^2^ patch. The amount of remaining viable adenovirus was measured at 1 week and then 1, 2, 3, and 6 months after incubation of samples (*n* = 6 per group per timepoint, except at 6 months when *n* = 4 per group). The % remaining in the sample was determined compared to time zero. No physical changes were observed in DMN stored at accelerated stability conditions. Points represent median, error bars represent inter-quartile range. ** *p* < 0.01, *** *p* < 0.001, compared to time zero, Kruskal–Wallis, one-way ANOVA, with Dunn’s multiple comparison test. (**B**) 25 °C, 60% RH, or (**C**) 40 °C, 75% RH, for 6 months.

**Figure 2 vaccines-09-00299-f002:**
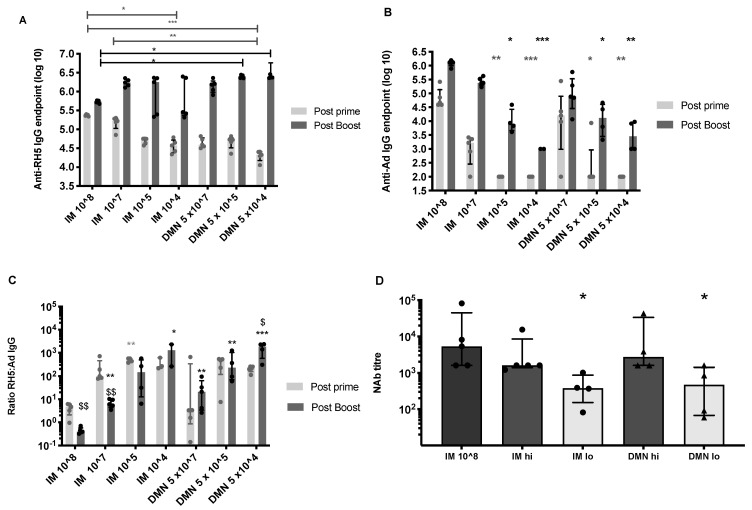
Effect of vaccine dose and route on the induction of IgG antibodies to PfRH5 antigen and to the adenovirus vector. Female Balb/c mice (*n* = 5 per group) were immunized on day 0 with AdHu5–PfRH5 by the IM route or using DMN patches. The following doses were administered by IM (from left to right); 1 × 10^8^ ifu, 1 × 10^7^ ifu, 1 × 10^5^ ifu, or 1 × 10^4^ ifu to each mouse. Alternatively, 5 × 10^7^ ifu, 5 × 10^5^ ifu, or 5 × 10^4^ ifu were administered in two patches to each mouse; one patch per ear. Serum IgG responses were assessed 8 weeks after the priming immunization (light grey bars). All mice were boosted on day 54 with 1 × 10^8^ ifu AdHu5–PfRH5 by the IM route and post-boost IgG responses were assessed 2 weeks after this boost (dark grey bars). (**A**) IgG responses to the encoded antigen PfRH5. * *p* < 0.05, ** *p* < 0.01, *** *p* < 0.001 by one-way ANOVA, Kruskal–Wallis test with Dunn’s multiple comparison. Light grey lines and stars indicate significant differences across post-prime samples, dark grey lines indicate significant differences across post-boost samples. (**B**) IgG responses to the AdHu5 vector. * *p* < 0.05, ** *p* < 0.01, *** *p* < 0.001 compared to control, IM 1 × 10^8^ ifu. (**C**) The ratio of anti-PfRH5 IgG titre to anti-vector IgG in each animal. * *p* < 0.05, ** *p* < 0.01, *** *p* < 0.001 compared to control, IM 1 × 10^8^ ifu. $ *p* < 0.05, $$ *p* < 0.01 of post-prime compared to post-boost within the same group. (**D**) Neutralizing antibody responses to AdHu5 vector 2 weeks after boosting. “IM lo” and “DMN lo” refers to priming mice with a low dose of 1–5 × 10^4^ ifu by IM injection or DMN patch, whereas “IM hi” and “DMN hi” refer to a high dose of 1–5 × 10^7^ ifu by IM or DMN patch-based administration in the prime * *p* < 0.05, compared to control, IM 1 × 10^8^ ifu. Bars represent median, error bars represent interquartile range, dots represent individual samples.

**Figure 3 vaccines-09-00299-f003:**
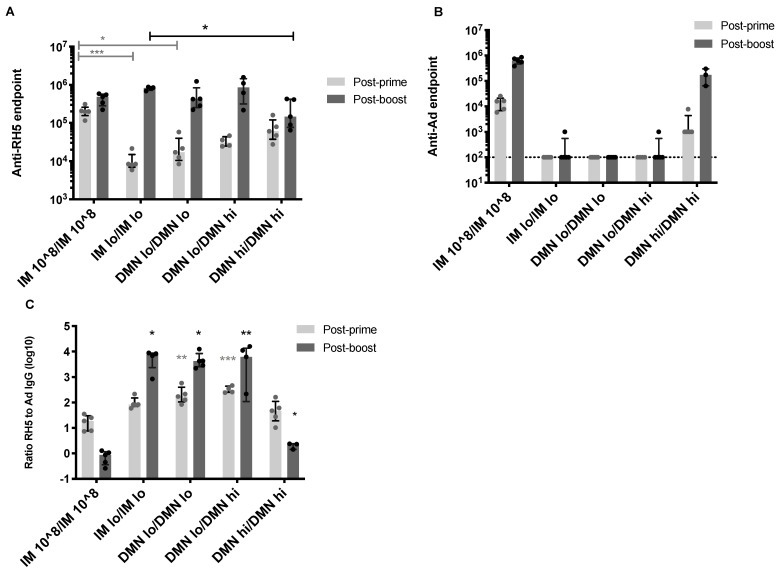
Antibody responses induced using a homologous dose and route in the prime and the boost. Female Balb/c mice (*n* = 5 per group) were immunized on day 0 with AdHu5–PfRh5 by the IM route or using DMN patches. The conventional priming and boosting by the IM route, using 1 × 10^8^ ifu acting as a control. Alternatively, mice were administered a **low dose of 5 × 10^4^ ifu** by IM injection or DMN patch (“IM lo” or “DMN lo”) or a **high dose of 5 × 10^7^ ifu by DMN patch (“DMN hi”)**. Serum IgG responses were assessed 8 weeks after the priming immunization. All mice were boosted on day 54 with a low or high dose of vaccine by the indicated route, and post-boost IgG responses were assessed 2 weeks after this boost. (**A**) IgG responses to the encoded antigen PfRH5. (**B**) IgG responses to the AdHu5 vector. (**C**) The ratio of the IgG titre to PfRH5 to the vector in each animal. * *p* < 0.05, ** *p* < 0.01, *** *p* < 0.001 by one-way ANOVA, Kruskal–Wallis test, with Dunn’s multiple comparison test compared to IM 10^8 in all graphs. Bars represent median, error bars represent interquartile range, circles represent individual samples.

**Figure 4 vaccines-09-00299-f004:**
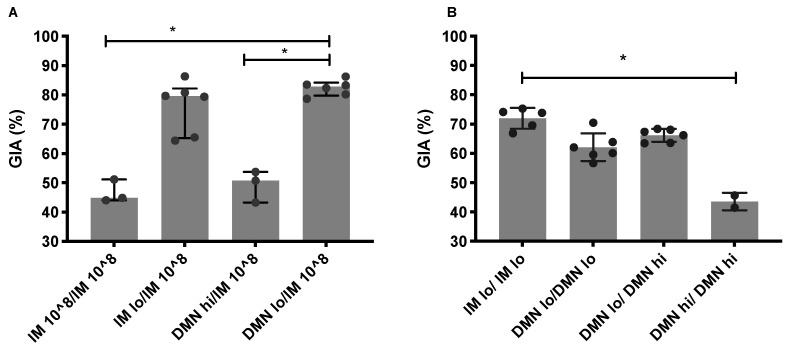
The dose and route of immunization impacts on functional antibody activity against malaria parasites. Female Balb/c mice (*n* = 10 per group) were immunized on day 0 with AdHu5–PfRH5 by the IM route or using DMN patches. The conventional priming and boosting by the IM route, used 1 × 10^8^ ifu and acted as a control regime. Alternatively, mice were administered a **low dose of 5 × 10^4^ ifu** by IM injection or DMN patch (“IM lo” or “DMN lo”) or a **high dose of 5 × 10^7^ ifu by DMN patch (“DMN hi”)**. All mice were boosted on day 54 with (**A**) the control dose of 1 × 10^8^ ifu by the IM route, or (**B**) with a low or high dose of vaccine by the indicated route. The level of growth inhibitory antibodies to *P. falciparum* 3D7 clone blood-stage parasites was assessed in purified IgG from serum taken 4 weeks post-boost and pooled into 2 to 6 samples, depending on volumes of available serum. * *p* < 0.05, by one-way ANOVA. Bars represent median, error bars represent interquartile range, circles represent individual samples.

**Figure 5 vaccines-09-00299-f005:**
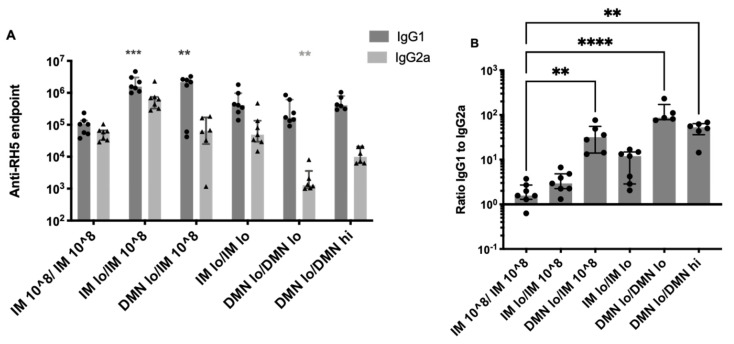
IgG subtypes induced by repeated immunization with low or high dose of vaccine by IM or skin route. Female Balb/c mice (*n* = 7 per group) were immunized on day 0 with AdHu5–PfRH5 by the IM route or using DMN patches. Mice were administered the standard 1 × 10^8^ ifu by IM injection twice (“IM 10^8/IM 10^8”). Alternatively, mice were administered a low dose of 5 × 10^4^ ifu by IM injection or DMN patch (“IM lo” or “DMN lo”) or a high dose of 5 × 10^7^ ifu by DMN patch (“DMN hi”). All mice were boosted on day 54 with a low or high dose or 1 × 10^8^ ifu of vaccine by the indicated route. (**A**) The endpoint titre of RH5-specific IgG1 (dark grey bars) and IgG2a (light grey bars) in serum taken at day 84 (4 weeks post-boost). (**B**) The ratio of IgG1 to IgG2a in each animal. Bars represent median, lines represent interquartile range, triangles and circles represent individual samples. ** *p* < 0.01; *** *p* < 0.0005, **** *p* < 0.0001 by Kruskal–Wallis one-way ANOVA comparing all groups to IM 10^8/IM 10^8.

**Figure 6 vaccines-09-00299-f006:**
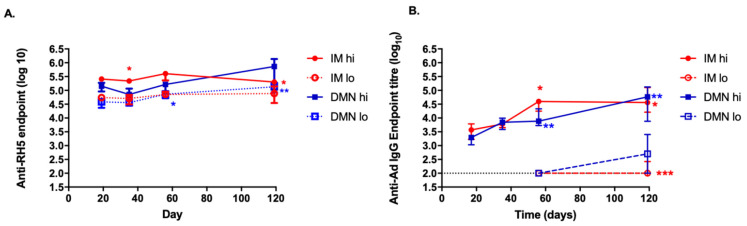
Effect of vaccine dose and route on the kinetics of IgG antibodies to PfRH5 antigen and to the adenovirus vector. Female Balb/c mice were immunized on day 0 with AdHu5–PfRH5 by the IM route or using DMN patches. Mice were administered the standard 1 × 10^8^ ifu by IM injection or with a DMN patch (“IM hi” or “DMN hi”). Alternatively, mice were administered a low dose of 5 × 10^4^ ifu by IM injection or DMN patch (“IM lo” or “DMN lo”). IgG responses to (**A**) the encoded antigen PfRH5 and (**B**) to the AdHu5 vector over time (*n* = 10 on day 17, *n* = 7 on days 35, 56, *n* = 4 on d119). Symbols represent median, bars represent interquartile range. For IgG responses to PfRH5, no significant differences in titres were observed over time for IM-lo or DMN-hi; for IM-hi; * *p* < 0.05 for day 17 compared to day 35 and day 119; * *p* < 0.05 for day 56 compared to day 119; for DMN-lo, * *p* < 0.05; day 17 compared to day 56; * *p* < 0.05 day 35 compared to day 119, ** *p* < 0.01; day 17 compared to day 119, by one-way ANOVA, Kruskal–Wallis test with Dunn’s multiple comparison. For IgG responses to AdHu5, in IM-hi; * *p* < 0.05 for day 17 compared to day 56 and day 119; in DMN-hi ** *p* < 0.01 for day 17 compared to day 56 and d119; in IM-lo; *** *p* < 0.001 compared to all other timepoints; no significant differences in titre were observed over time for DMN-lo, by one-way ANOVA, Kruskal–Wallis test with Dunn’s multiple comparison.

## Data Availability

The data presented in this study are openly available in FigShare at https://doi.org/10.6084/m9.figshare.14198867.v1.

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
