# Peer review of "Low Adenovirus Vaccine Doses Administered to Skin Using Microneedle Patches Induce Better Functional Antibody Immunogenicity as Compared to Systemic Injection"

_vaccines, 2021, doi:10.3390/vaccines9030299_

Round 1
Reviewer 1 Report
MDPI-Vaccines 2021-01-29
The manuscript ID: Vaccines: 1099553 on “Low adenovirus vaccine doses administered to skin using microneedle patches induce better functional antibody immunogenicity as compared to systemic injection” by Olivia Flynn from Dr. Moore AC’s lab. Authors reported that a new method, dissolvable microneedle patches (DMN) for using in evaluation of Adenovirus vector based malaria vaccine, as comparison of conventional intramuscular injection (MI).
DMN and IM is two different way to inoculate the vaccine, thus antigen- AdHu5-PfRH5 via IM would be much quick than via DMN method. Induced IgG antibody against Ad5 vector will appear early than DMN. The time point for antibody evaluation is at the day 54 after vaccination.
Question 1, Fig.1 A is hard to demonstrate the results, due to limited time point for the liquid 40°C and DMN 25°C, relative more time points with DMN 40°C. The results with the liquid 40°C after 6 month lacks. Thus it could not support the authors’ conclusion.
Question 2, Authors could indicate how quickly distribution of the Adv-PrRH5 via DMN method in vivo, in order to demonstrate the antibody induction simultaneously as the conventional IM. If DMN takes few days or weeks more than IM method, it will be better to have two time points such as at the day 30 and the day 54, and compare the antibody titter and Nab titter.
Question 3: Fig 2 and Fig 3 should be modified to be clear and better, since it is hardly to see how many mice in each group.
Question 4. Fig 4. the number of mouse in the important group: IM 108 /IM108, DMN 108 /IM108, and DMNhi /DMN hi are much less than other groups, it seems that there are only two mice in DMNhi /DMN hi group so that the reliability and conclusion could be questionable. Fig 4 lacks “A” or “B” to each subfigure. In Fig.4 No statistically significance in Fig.4B.
DMN could be used in human?
Author Response
Response to Reviewer 1:
We thank the Reviewer for their considered comments and feedback.
Question 1, Fig.1 A is hard to demonstrate the results, due to limited time point for the liquid 40°C and DMN 25°C, relative more time points with DMN 40°C. The results with the liquid 40°C after 6 month lacks. Thus it could not support the authors’ conclusion.
As outlined in the results (Section 3.1) “Adenovirus stored in the liquid form at the higher temperature, quickly degraded within a week and viable virus was undetectable at one month.” Therefore it was impossible to generate results beyond a 1 month period.
In addition to short-term stability of an AdHu5 vector embedded in our DMN patches and kept at ambient temperature over 14 days (reference 15), we also determined that AdHu5 in DMN patches were stable for 1 month at 25°C and 60% relative humidity (RH) in a preliminary stability study of patches (data not shown). Therefore, when we performed this larger stability study at multiple conditions and time-points, the main objectives were (i) to extend the stability incubation time period of patches held at 25°C and 60% RH and (ii) to further stress the stability of patches by increasing the temperature and relative humidity conditions. The study was also established in agreement with the testing frequency outlined in ICH Q1A guidelines (reference 3); this guidance suggests that at accelerated storage (25°C for a product intended to be stored in a refrigerator), a minimum of three time points, including initial and final time points (e.g., 0, 3, and 6 months), from a 6-month study should be examined. Increased time points were examined at the 40°C and 75% RH as we hypothesised that degradation would occur early in these harsh conditions and we aimed to identify the time window when this degradation happened. The results demonstrate that the vaccine is stable in our DMN patches up to 8 weeks at these harsh conditions of 40°C and 75% RH.
Question 2, Authors could indicate how quickly distribution of the Adv-PrRH5 via DMN method in vivo, in order to demonstrate the antibody induction simultaneously as the conventional IM. If DMN takes few days or weeks more than IM method, it will be better to have two time points such as at the day 30 and the day 54, and compare the antibody titter and Nab titter.
Performing pharmacokinetic studies to examine the speed of distribution of vaccines in vivo is very rarely conducted and cannot be correlated with the kinetics of antibody induction. We know that our patches deliver vaccine into skin in <10 minutes (reference 15). We examined the NAb titer only after boosting.
We have now included a study that examined the kinetics of the IgG responses to the antigen and to the vector after a single immunization (Fig. 6). We believe that this demonstrates that although the dose of vaccine affected the magnitude of the response, no differences in the kinetic profile are observed when vaccine is administered with a DMN patch or by IM injection. We amended the Results and Discussion section to include details on this study.
Question 3: Fig 2 and Fig 3 should be modified to be clear and better, since it is hardly to see how many mice in each group.
As stated in the figure legend in Figures 2 and 3, there were 5 mice in each group. Due to low variability within each group, the points representing individual mice in some groups overlap. However, we believe that is also important to demonstrate this consistency of response. If the reviewer can suggest an alternative method then we will be happy to consider its use.
Question 4. Fig 4. the number of mouse in the important group: IM 108 /IM108, DMN 108 /IM108, and DMNhi /DMN hi are much less than other groups, it seems that there are only two mice in DMNhi /DMN hi group so that the reliability and conclusion could be questionable. Fig 4 lacks “A” or “B” to each subfigure. In Fig.4 No statistically significance in Fig.4B.
As stated in the figure legend of Figure 4, 10 mice were immunized in each group. In order to conduct this GIA assay, IgG must be purified from serum, pre-adsorbed with uninfected human erythrocytes, sterile filtered, heat inactivated and normalised for IgG concentration to 1mg/ml (according to protocols outlined in references 33 and 34). As low volumes of serum are recovered from mice, individual serum samples must be pooled so that a sufficient volume of purified IgG is available to conduct the GIA assay. The number of pooled samples in each group was dependent on the initial volumes of serum recovered from each mouse and the yield of purified IgG. Thus, in the DMNhi/DMNhi group, only 2 pools could be created. However, these 2 pools are representative of the initial 10 mice; as can be seen from the figure, the 2 pools had similar GIA activity.
We have corrected the text “pooled into 3 or 6 samples” to “pooled into 2 to 6 samples”.
We have included “A” and “B” in the figure and included markers of statistical differences in Fig. 4B.
DMN could be used in human?
DMN patches to deliver vaccines have been used in several species, including humans. We have successfully administered vaccines to pigs (reference 54 in the original text, for example), others have administered them to macaques (reference 25 for example). They have also been tested in clinical trials for vaccine administration; this has been included in the Discussion (line 424).
Reviewer 2 Report
Flynn et al. studied the “Low adenovirus vaccine doses administered to skin using microneedle patches induce better functional antibody immunogenicity as compared to systemic injection”.
In this study, the authors have prepared the novel dissolvable microneedle (DMN) patches for skin immunization. In addition, the authors have compared the skin immunization of DMN patches to conventional intramuscular (IM) immunization with low and high doses; vice-versa. As authors anticipated, the novel DMN patches overpowered the antigen-specific immune responses. Furthermore, despite less immunity against adenovirus vector, this approach offered high stability for vaccines, which is very much essential for break the cold chain. Data is well presented. The discussion is well placed.
Some micro corrections are mandatory before publishing.
Line 21: “so using” is redundant in the sentence.
Acronym “DMN” should be used once abbreviated, which is lacking throughout the manuscript.
Section 2.4: An illustration that explains the immunization schedule would be useful for readers and allow other researchers to follow.
- falciparum (Line 154 & 160) and in vitro (Line 154) should be italics.
Figure 5: Correct the x-axis label “IM lo/IM hi” ------ “IM lo/IM 10^8”
Figure 5B: Statistics have not represented, although they look significant.
In the discussion, please place a reasonable discussion on the delivery of vaccines through microneeds, antigen release and kinetics.
Line 419: Expand “PLGA”
Line 429: For example, repeated IM use of an adenovirus type 35 by the IM route ------------ For example, repeated use of an adenovirus type 35 by the IM route
Please check the reference format: 23, 44, 49, 57
Author Response
Response to Reviewer 2:
We thank the Reviewer for their considered comments and feedback.
Line 21: “so using” is redundant in the sentence.
This phrase “.. more so using”; demonstrates that the DMN patch was overall, better than the IM route. As the Reviewer has indicated that they do not feel qualified to judge about the English language and style, we propose to leave this statement as currently written.
Acronym “DMN” should be used once abbreviated, which is lacking throughout the manuscript.
We are unsure what the Reviewer means in this sentence.
Section 2.4: An illustration that explains the immunization schedule would be useful for readers and allow other researchers to follow.
We have included Supplementary Figure 1 to explain the immunisation schedule.
- falciparum (Line 154 & 160) and in vitro (Line 154) should be italics.
These have been corrected.
Figure 5: Correct the x-axis label “IM lo/IM hi” ------ “IM lo/IM 10^8”
Figure 5B: Statistics have not represented, although they look significant.
These have been corrected.
In the discussion, please place a reasonable discussion on the delivery of vaccines through microneeds, antigen release and kinetics.
We have included a discussion on the kinetic profile of the antibody response in lines 460-467.
Line 419: Expand “PLGA”
This has been expanded.
Line 429: For example, repeated IM use of an adenovirus type 35 by the IM route ------------ For example, repeated use of an adenovirus type 35 by the IM route
This has been corrected.
Please check the reference format: 23, 44, 49, 57
All of the references have been checked and conform to the journal guidelines.
Round 2
Reviewer 1 Report
Line 376: 1X108 ifu; 5X104 ifu: should collect to 1X108 ifu; 5X104 ifu.